# Comparison of Simulated Multispectral Reflectance among Four Sensors in Land Cover Classification

Feng Chen [1,2], Wenhao Zhang [1], Yuejun Song [3], Lin Liu [4] and Chenxing Wang [5,*]

1. College of Computer and Information Engineering, Xiamen University of Technology, Xiamen 361024, China; chenfeng@xmu.edu.cn (F.C.)
2. Big Data Institute of Digital Natural Disaster Monitoring in Fujian, Xiamen University of Technology, Xiamen 361024, China
3. Jiangxi Key Laboratory of Soil Erosion and Prevention, Jiangxi Academy of Water Science and Engineering, Nanchang 330029, China
4. School of Physics, Huazhong University of Science and Technology, Wuhan 430074, China
5. State Key Laboratory of Urban and Regional Ecology, Research Center for Eco-Environmental Sciences, Chinese Academy of Sciences, Beijing 100085, China
* Correspondence: cxwang@rcees.ac.cn

**Abstract:** Multispectral images accessible free of charge have increased significantly from the acquisitions by the wide-field-of-view (WFV) sensors onboard Gaofen-1/-6 (GF-1/-6), the Operational Land Imager (OLI) onboard Landsat 8 (L8), and the Multi-Spectral Instrument (MSI) onboard Sentinel-2 (S2). These images with medium spatial resolutions are beneficial for land-cover mapping to monitor local to global surface dynamics. Comparative analyses of the four sensors in classification were made under different scenarios with five classifiers, mainly based on the simulated multispectral reflectance from well-processed hyperspectral data. With channel reflectance, differences in classification between the L8 OLI and the S2 MSI were generally dependent on the classifier considered, although the two sensors performed similarly. Meanwhile, without channels over the shortwave infrared region, the GF-1/-6 WFVs showed inferior performances. With channel reflectance, the support vector machine (SVM) with Gaussian kernel generally outperformed other classifiers. With the SVM, on average, the GF-1/-6 WFVs and the L8 OLI had great increases (more than 15%) in overall accuracy relative to using the maximum likelihood classifier (MLC), whereas the overall accuracy improvement was about 13% for the S2 MSI. Both SVM and random forest (RF) had greater overall accuracy, which partially solved the problems of imperfect channel settings. However, under the scenario with a small number of training samples, for the GF-1/-6 WFVs, the MLC showed approximate or even better performance compared to RF. Since several factors possibly influence a classifier's performance, attention should be paid to a comparison and selection of methods. These findings were based on the simulated multispectral reflectance with focusing on spectral channel (i.e., number of channels, spectral range of the channel, and spectral response function), whereas spatial resolution and radiometric quantization were not considered. Furthermore, a limitation of this paper was largely associated with the limited spatial coverage. More case studies should be carried out with real images over areas with different geographical and environmental backgrounds. To improve the comparability in classification among different sensors, further investigations are definitely required.

**Keywords:** Gaofen; WFV; multispectral; Landsat 8; Sentinel-2; machine learning; random forest; land cover

## 1. Introduction

Since it has a unique capability to observe large areas in an efficient manner, satellite remote sensing has played an essential role in numerous applications, such as in land-cover mapping and change detection [1]. Land-cover mapping with medium spatial resolutions is critical for monitoring local to global surface dynamics [2], which further supports

informed decision making. Increases in images that are freely accessible with no charge have been accelerating land-cover production [3], as well as method development in data processing [4]. Currently, in addition to the data products of Landsat and Sentinel, the images acquired by the wide-field-of-view (WFV) sensors of Gaofen-1 (GF-1) and Gaofen-6 (GF-6) have been accessible free of charge (http://www.cnsa.gov.cn/n6758823/n6758838 /c6808018/content.html, accessed on 15 April 2023).

Considering multispectral imagery with medium spatial resolutions, pixel-based classification has been widely applied for regional and global land-cover mapping [1,3,5–7]. For the classification at medium spatial resolution, characterization of spectral properties (i.e., channel reflectance) is of highest importance compared to metrics based on time series observations and other ancillary information [8,9]. Differences in spectral channels are always observable between different sensors. Consequently, the comparability in surface mapping between sensors deserves investigation.

In terms of the GF-1/-6 WFVs, the Landsat 8 Operational Land Imager (L8 OLI), and the Sentinel-2 Multi-Spectral Instrument (S2 MSI), many efforts have been made to address the comparability issue mainly concerning two different sensors [10–15]. Both the GF-1 WFV and the GF-6 WFV slightly outperformed the L8 OLI in land-cover classification [11,13]. Thanks to its red-edge (RE) channels, the GF-6 WFV had better separability for arable land, forestland, grassland and shrub over an area with complex terrain, thereby obtaining a greater accuracy than the L8 OLI [11]. Since it has advantages both in spectral channel and in spatial resolution, the S2 MSI outperformed the L8 OLI in many cases, such as in forest variable prediction [10] and in built-up land mapping [14]. Using a simulated data collection, the S2 MSI produced a 7% greater overall accuracy than L8 OLI in discriminating grass under different conditions [12]. At the same time, comparisons of the three sensors have been discussed in a few investigations [16–20]. In these cases, the S2 MSI generally showed advantages over other sensors. For example, for crop type classification, the S2 MSI obtained the highest accuracy, slightly outperforming the L8 OLI, while the GF-1 WFV had the lowest accuracy [16]. In the extraction of tidal creeks, the L8 OLI and the GF-1 WFV images produced similar results, while the S2 MSI images produced a greater number of the tidal creeks with more levels [18]. In forest/non-forest extraction, the S2 MSI outperformed both the GF-1 WFV and the L8 OLI [20]. These comparative analyses highlighted the differences between sensors in spectral channel (e.g., spectral response function) and in spatial resolution, causing classification inconsistency. Therefore, these differences should be considered in a synergistic application of different sensors for land surface mapping [14].

To our knowledge, comprehensive investigations of the comparability in classification among these multispectral sensors (i.e., the GF-1 WFV, the GF-6 WFV, the L8 OLI, and the S2 MSI) are still relatively limited. Meanwhile, random forest (RF) was mostly used in the comparisons for land-use/cover classification [11,16,19,20], whereas other classifiers were rarely discussed. This study comprehensively investigated the characterization and comparison of the four multispectral sensors, specifically in terms of classification and channel reflectance (Figure 1). Since paired observations synchronously acquired by the four sensors were inaccessible, a data collection of broadband channel reflectance was generated from well-prepared hyperspectral records of the Airborne Visible Infrared Imaging Spectrometer (AVIRIS), with a focus on spectral differences. In addition to RF, other methods were implemented in the comparisons, including maximum likelihood classifier (MLC), K-nearest neighbor (KNN), decision tree (DT), and support vector machine (SVM). Nevertheless, this investigation did not aim to identify or propose the classifier(s) most suitable for land-cover/use mapping, instead focusing on the comparability issue among the four sensors. Generally, the objectives were to (1) figure out the comparability problem in classification among the four sensors, and (2) discuss possible factors affecting classification comparison. As the simulated multispectral reflectance was used, impacts associated with spectral channel properties (i.e. number of channels, spectral range of the channel, and spectral response function) were mainly considered. However, in fact, there

are many factors influencing classification results in practice [21]. The paper is organized as follows: Section 2 introduces the data used and the methods adopted; Section 3 presents the comparison results in sensor characteristics, channel reflectance, and classification; Section 4 provides a discussion, mainly on classifier importance and limitations, as well as the contribution of individual channels to classification; Section 5 provides conclusions.

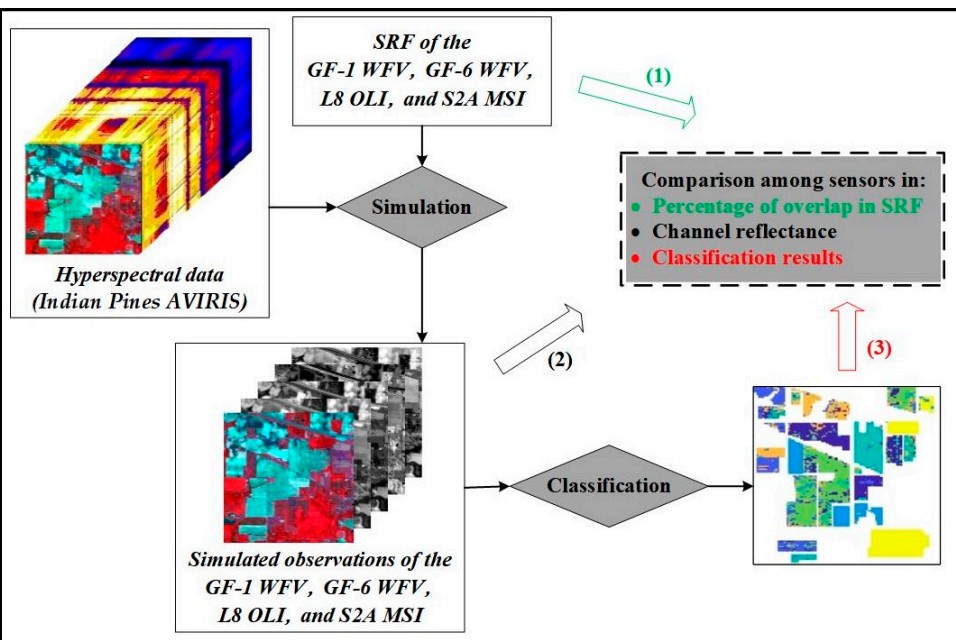

**Figure 1.** Flowchart presenting the major contents in this study. Comparisons among sensors are conducted in terms of (1) percentage overlap in spectral response function (SRF), (2) channel reflectance, and (3) classification.

## 2. Materials and Methods

### 2.1. Four Sensors in Comparison

GF-1 and GF-6 were launched on 26 April 2013 and on 2 June 2018, respectively. The GF-6 WFV is similar to the GF-1 WFV in spatial resolution and image swath, i.e., at 16 m spatial resolution with a wide swath (about 800 km), but has four additional channels (Table 1). A constellation network of GF-1 and GF-6 was developed and has been in operation since August 2018. The China National Space Administration announced an open data policy to share global imagery acquired by the GF-1/-6 WFVs at the GEO Week 2019 Plenary. Accordingly, registered users around the world are able to obtain the WFV data freely through the data platform (CNSA-GEO) (http://www.cnsa.gov.cn/n67588 23/n6758838/c6808018/content.html, accessed on 15 April 2023). The Landsat program has continuously observed the Earth's land surface since 1972. The L8 OLI, launched on 11 February 2013, has eight channels at 30 m spatial resolution covering the visible, near-infrared (NIR), and shortwave infrared (SWIR) regions, along with a panchromatic (PAN) channel (Table 1). Meanwhile, L9 launched on 27 September 2021, as the newest member of the Landsat program, is equipped with the sensors as improved replicas of those onboard L8. Sentinel-2 currently comprises two satellites (called S2A and S2B) equipped with the MSI. S2A and S2B were launched on 23 June 2015 and on 7 March 2017, respectively. The MSI has 13 channels with medium spatial resolutions (10 m to 60 m) (Table 1). Compared with the L8 OLI, the S2 MSI has three channels within the RE wavelengths. There are slight differences in the MSI between S2A and S2B for several channels. In this study, the S2A MSI was considered in the discussion (Figure 2). Table 1 presents general information about these sensors. There are four WFV sensors onboard GF-1 with minor differences [22], of which the WFV1 is mainly discussed in this paper.

**Table 1.** General information about the channels of the GF-1 WFV, the GF-6 WFV, the L8 OLI, and the S2 MSI.

| GF-1 WFV | GF-6 WFV | L8 OLI | S2 MSI |
|---|---|---|---|
| — | 7 Costal: 400–450 nm | 1 CA: 433–453 nm | 1 CA: 433–453 nm (60 m) [1] |
| 1 Blue: 450–520 nm | 1 Blue: 450–520 nm | 2 Blue: 450–515 nm | 2 Blue: 458–523 nm (10 m) |
| 2 Green: 520–590 nm | 2 Green: 520–590 nm | 3 Green: 525–600 nm | 3 Green: 543–578 nm (10 m) |
| 3 Red: 630–690 nm | 3 Red: 630–690 nm | 4 Red: 630–680 nm | 4 Red: 650–680 nm (10 m) |
| 4 NIR: 770–890 nm | 4 NIR: 770–890 nm | 5 NIR: 845–885 nm | 8 NIR: 785–900 nm (10 m) |
| | | | 8a NIR: 855–875 nm (20 m) |
| — | 5 RE1: 690–730 nm | — | 5 RE1: 694–713 nm (20 m) |
| | 6 RE2: 730–770 nm | | 6 RE2: 713–749 nm (20 m) |
| | | | 7 RE3: 769–797 nm (20 m) |
| — | — | 6 SWIR1: 1560–1660 nm | 11 SWIR1: 1565–1655 nm (20 m) |
| | | 7 SWIR2: 2100–2300 nm | 12 SWIR2: 2100–2280 nm (20 m) |
| — | 8 Yellow: 590–630 nm | — | 9 WV: 932–958 nm (60 m) |
| — | — | 9 Cirrus: 1360–1390 nm | 10 Cirrus: 1337–1412 nm (60 m) |
| — | — | PAN: 500–680 nm (15 m) [2] | — |

[1] Values in parentheses represent the spatial resolution for a channel of the S2 MSI. The spatial resolution for all channels of the GF-1/-6 WFVs is 16 m, and that for all channels except the PAN of the L8 OLI is 30 m. [2] The spatial resolution for the L8 OLI PAN is 15 m. In addition to ordinary channels within visible wavelengths, near-infrared wavelengths, and shortwave infrared wavelengths, the channels for costal aerosol (CA) and water vapor (WV), as well as those within the red-edge (RE) region, are included.

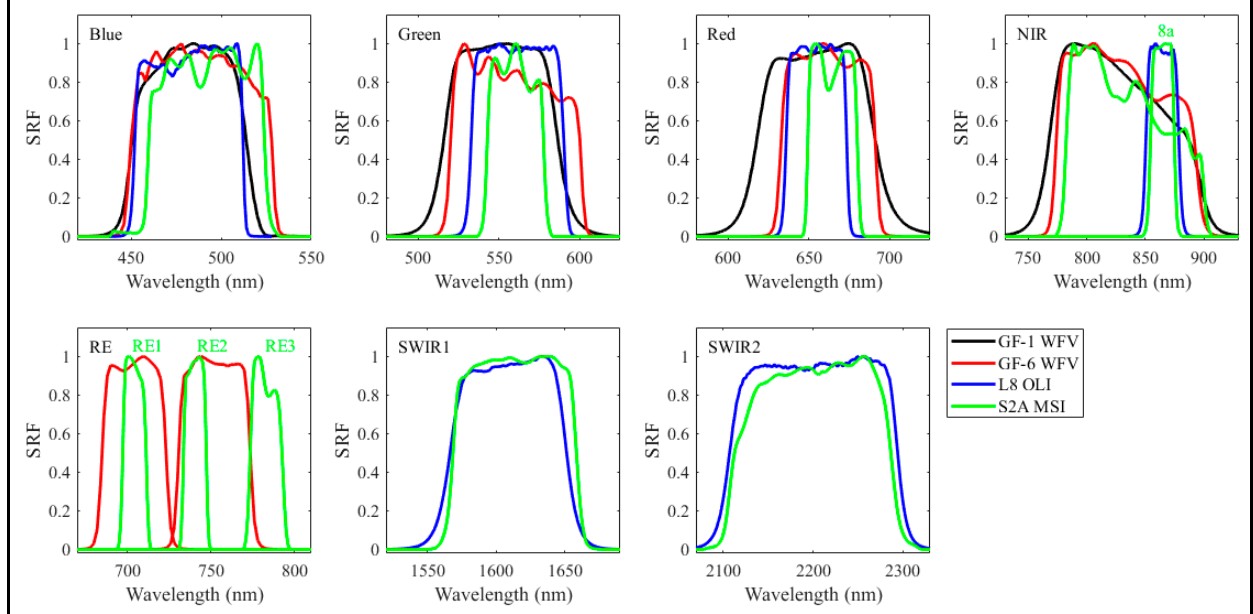

**Figure 2.** Spectral response functions (SRFs) for corresponding channels of the four multispectral sensors, including the GF-1 WFV (with blue line), the GF-6 WFV (with black line), the L8 OLI (with green line), and the S2A MSI (with red line). "8a" in the subplot indicates the 8a NIR channel of the S2A MSI. The GF-1 WFV1 is presented in this figure.

### 2.2. Spectral Response Function

The spectral response function (SRF) is an important aspect of sensor settings [23,24]. Due mainly to the differences in SRF, channel reflectance and its derived variables (e.g., spectral indices and feature components through transformation) usually differ among sensors [25,26]. Accordingly, the SRF differences need to be quantified and compensated for, especially when observations acquired from different sensors are used jointly [14,23]. The comparability issues among four sensors (i.e., the GF-1 WFV, the GF-6 WFV, the L8 OLI, and the S2A MSI) are firstly discussed, including SRF and channel reflectance (see Section 2.3).

The SRFs for the GF-1 WFV and the GF-6 WFV were provided by China Center for Resources Satellite Data and Application (CRESDA, https://www.cresda.com, accessed on 1 June 2022). The SRFs for two WFV sensors were sampled at 1 nm steps within 400–1000 nm. The SRFs for the L8 OLI were accessed at https://landsat.usgs.gov/spectral-characteristics-viewer (accessed on 1 June 2022), while the SRFs for the S2A MSI were obtained from https://sentinel.esa.int/web/sentinel/user-guides/sentinel-2-msi/document-library (accessed on 1 June 2022). The SRFs for the L8 OLI and the S2A MSI were sampled at 1 nm. As shown in Figure 2, the SRFs of a specific channel (nominally the same one) varied more or less among the four sensors, and the inter-sensor difference depended on the channel considered. To compare specific channel pairs quantitatively, the percentage overlap in SRF (Equation (1)) was calculated, as described in [22,27]. The channels within the visible, NIR, SWIR, and RE regions are mainly discussed in this investigation, whereas others are not considered including the coastal (aerosol), cirrus, WV, and PAN of the L8 OLI (Table 1).

$$PO_i(A, B) = \frac{SRF_i^A \cap SRF_i^B}{SRF_i^B} \times 100\%, \tag{1}$$

where $SRF_i^A$ and $SRF_i^B$ are the spectral response functions for channel i of sensor A and sensor B, respectively. Accordingly, $PO_i(A, B)$ indicates the percentage overlap for sensor A within sensor B, representing the part of sensor A contained in sensor B.

### 2.3. Simulation of Multispectral Observations from Hyperspectral Data

Because it was impossible to obtain observations from the four sensors synchronously (i.e., the GF-1 WFV, the GF-6 WFV, the L8 OLI, and the S2A MSI), multispectral images simulated from hyperspectral data were used. This was considered a feasible way to simulate multispectral observations from hyperspectral data, in comparing sensors with broadband channels [12,25,26]. To compare classification among sensors, the simulated channel reflectance from well-processed hyperspectral data allows focusing mainly on spectral differences [28]. As in previous investigations [29], the Indian Pines AVIRIS data [30] were used for the multispectral image simulation. The Indian Pines AVIRIS data were well calibrated, along with detailed information about ground truth; consequently, they have been widely used as a benchmark [29,31]. The Indian Pines AVIRIS data were acquired on 12 June 1992, covering approximately a 2.9 × 2.9 km$^2$ area in Tippecanoe County, Indiana, USA, with a size of 145 × 145 pixels. Meanwhile, a total of 200 channels were used in the simulation of multispectral radiance after removing 20 spectral channels (including 104–108, 150–163, and 220) due mainly to noise and atmospheric water absorption. Information about the calibrated AVIRIS channels can be found at https://engineering.purdue.edu/~biehl/MultiSpec/aviris_documentation.html (accessed on 1 June 2022).

The main procedures of simulation, including (1) obtaining synthesized channel radiance and (2) estimating the channel reflectance considering solar irradiance (Table 2), were implemented to generate broadband channel reflectance form the well-calibrated AVIRIS data. Details on these procedures are presented in [29]. In addition, to retrieve channel reflectance from simulated radiance, the exo-atmospheric solar irradiance (ESUN) is required to compensate for solar irradiance. In this paper, ESUN values based on the Thuillier solar spectrum profile [32] were used (Table 2). The Thuillier solar spectrum profile has been used for the ESUN values of the Landsat sensors [33] and for the S2 MSI [34]. Detailed processes for the ESUN estimation with solar spectrum profile were discussed [29]. The sensors' radiometric properties (e.g., dynamic ranges of radiance and quantization level) were not considered in the simulation of channel radiance, due mainly to the inability to access this kind of information for the GF-1/-6 WFVs. However, considering the Indian Pines AVIRIS data, quantization effects on simulated multispectral reflectance (e.g., simulation of the L8 OLI) differed among channels, in which significant effects over the SWIR region were observed [35].

**Table 2.** Exo-atmospheric solar irradiance (ESUN) values (in $W \cdot m^{-2} \cdot \mu m^{-1}$) for sensors and channels mainly discussed, estimated using the Thuillier solar spectrum.

| Channel | GF-1 WFV1 | GF-6 WFV | L8 OLI | S2A MSI |
|---------|-----------|----------|--------|---------|
| Blue | 1997 [1968.63] [1] | 1972 [1952.16] | 2005 | 1960 |
| Green | 1820 [1849.19] | 1817 [1847.43] | 1821 | 1823 |
| Red | 1548 [1571.46] | 1534 [1554.76] | 1550 | 1512 |
| NIR | 1064 [1079] | 1057 [1074.06] | 952.0 | 1042 |
| 8a NIR | — | — | — | 955.0 |
| RE1 | — | 1422 [1412] | — | 1425 |
| RE2 | — | 1274 [1267.39] | — | 1288 |
| *RE3* | — | — | — | *1162* |
| SWIR1 | — | — | 247.6 | 245.6 |
| SWIR2 | — | — | 85.46 | 85.25 |
| Yellow | — | 1700 [1736.92] | — | — |

[1] The ESUNs for the GF-1/-6 WFVs in square brackets are publicly available at http://www.cresda.com (accessed on 1 June 2022), but without any information on the solar spectrum used for the estimation of these ESUNs. The Thuillier-based ESUNs used in this investigation were corrected to four significant digits, while the published values for the GF-1/-6 WFVs were kept as the original ones.

Further comparisons in reflectance among the four sensors were mainly performed for channels within the visible (i.e. blue, green, and red), NIR, RE, and SWIR wavelengths (Table 1). The median relative difference (MdRD) was used in the comparative analyses, as described in previous investigations [25,26].

$$MdRD_i(A, B) = median \left[ \frac{2 \times \left( Ref_{ij}^B - Ref_{ij}^A \right)}{\left( Ref_{ij}^A + Ref_{ij}^B \right)} \times 100\% \right], \tag{2}$$

where $Ref_{ij}^A$ and $Ref_{ij}^B$ are reflectance values of sample j (j = 1, 2, ... , n) for channel i of sensor A and sensor B respectively, and $median[]$ describes the operation to obtain the median difference. The $MdRD_i(A, B)$ measures the median relative difference for channel i between sensor A and sensor B, using sensor A as reference. As normalized for the differences in channel reflectance, the median relative difference makes comparisons among different channels more feasible and easier [26].

*2.4. Comparisons in Classification*

Five methods widely used in pixel-based classification were implemented and compared, including MLC, KNN, DT, RF, and SVM. The MLC as a parametric classifier has been popular in practice, and it has also served as a baseline in classification comparisons [31,36]. KNN and DT both are nonparametric methods, which are easily applied with a few parameters required [29,31]. In particular, to perform the KNN in classification, only the number of nearest neighbors surrounding a target pixel is predefined necessarily. In this study, for each classification case, the number of nearest neighbors was determined according to the training tests in advance. Specifically, a sequence of training tests was performed with different numbers of nearest neighbors (3–50) for each case, from which that with the minimum loss was selected and used in classification. At the same time, DT is usually viewed as a set of if–then rules, being regarded as an intuitively simple classifier [31]. The hyperparameters of DT were automatically determined through an optimizing procedure in each classification case, in this study. However, there are usually problems for the DT, such as generating a nonoptimal solution and overfitting [31]. In contrast, RF is an ensemble classifier using many decision trees to tackle the shortcomings of a single DT [37], through which the error in classification can be reduced. As stated in a recent review [38], the RF is relatively easily implementable with the capability to tackle problems (e.g., in classification and regression) with a small number of training samples, showing the ability to generate competitive results with other complex models (e.g., convolutional neural networks). Because of its insensitivity to the noise of the target

data and simplicity for application, the RF has been popularly adopted in land-use/cover mapping [4,7,39–42]. To train an RF model, the major parameters required are the number of decision trees and the minimum number of leaves. In this study, an RF model (named TreeBagger) embedded in MATLAB (R2017) which supports bootstrap aggregation was implemented. Its number of decision trees and minimum number of leaves were set to 100 and three, respectively, determined according to a series of tests (see Section 4.3). In addition, taking advantage of its bagging strategy in the training process, the RF has been used to measure the importance of individual variables and to select significant features for classification [29,31,43]. The features selected through RF are generally consistent with existing domain knowledge [44]. Accordingly, the RF was used to test variable (i.e., channel reflectance) importance (see Section 4.2), as well as in classification. As a nonparametric classifier, SVM aims to find the optimal boundary maximizing the separation between training samples [31]; it has been widely applied, especially in land-use classification [38]. In particular, the SVM is able to produce results with higher accuracy even with small training samples [31,38]. According to previous experiments [29], the SVM with Gaussian kernel was implemented.

Seven classes with relatively larger samples (>500 pixels) were selected for the comparative analyses in classification, although there were 16 classes labeled within the ground truth map for the Indian Pines AVIRIS data [30]. The classes selected were corn-notill (Cn) with 1428 samples, corn-mintill (Cm) with 830 samples, grass-trees (Gt) with 730 samples, soybean-notill (Sn) with 972 samples, soybean-mintill (Sm) with 2455 samples, soybean-clean (Sc) with 593 samples, and woods (Ws) with 1265 samples. To train a classifier properly, training samples are critical. For each class, the number of training samples is recommended to be greater than 10–30 times the number of variables (e.g., channels) [45]. The principle of stratified random sampling was implemented to select training samples. Specifically, the same number of samples for each class was randomly selected in each case. Furthermore, to understand the performances of classifier related to training samples, a series of tests were conducted with certain training samples for each class under four scenarios, including with 50 training samples, with 100 training samples, with 200 training samples, and with 300 training samples. Under each scenario, 20 data collections of randomly selected samples containing channel reflectance (Table 3) and the corresponding ground truth were used to separately train the classifiers. Under each scenario, 20 classification cases were obtained for a specific sensor with one classifier. In each classification case, the samples not selected for training were used for validation in accuracy assessment. The overall accuracy (OA) and Kappa coefficient were adopted in general assessment. In addition, as in the companion study [29], the producer's accuracy (PA), user's accuracy (UA), and the harmonic average (also known as the F1-score) were used in assessing individual classes.

**Table 3.** Inputs and methods for different classification analyses.

| | Sensor | Channels Used | Classifiers Used |
|---|---|---|---|
| Test 1 [1] | GF-1 WFV | Blue, green, red, NIR | MLC, DT, KNN, RF, SVM |
| | GF-6 WFV | Blue, green, red, NIR, RE1, RE2 | MLC, DT, KNN, RF, SVM |
| | L8 OLI | Blue, green, red, NIR, SWIR1, SWIR2 | MLC, DT, KNN, RF, SVM |
| | S2A MSI | Blue, green, red, NIR, 8a NIR, RE1, RE2, SWIR1, SWIR2 | MLC, DT, KNN, RF, SVM |
| Test 2 [2] | GF-1 WFV | Blue, green, red, NIR + SWIRs of the L8 OLI | MLC, DT, KNN, RF, SVM |
| Test 3 [3] | L8 OLI | Blue, green, red, NIR | MLC, DT, KNN, RF, SVM |

[1] Test 1 was intended to compare the four multispectral sensors in classification, and to show the performance differences among classifiers, serving as main topics in this study; [2] Test 2 was to show the importance of the SWIR channels, through adding the SWIRs of the L8 OLI to the GF-1 WFV; [3] Test 3 was conducted using visible and NIR channels of the L8 OLI as inputs (i.e., without two SWIRs) to show the importance of the SWIR channels. Information about the channels is presented in Table 1.

The channel reflectance was mainly used as input for individual classifiers. All channels of the GF-1 WFV (i.e., Blue, green, red, and NIR) and six channels of the GF-6

WFV (i.e., blue, green, red, and NIR, as well as two REs) were considered. Meanwhile, six channels of the L8 OLI (i.e., blue, green, red, NIR, and two SWIRs) and nine channels of the S2A MSI (i.e., blue, green, red, two NIRs, two REs, and two SWIRs) were implemented in classification (see Test 1 in Table 3). Additionally, more tests were conducted to show the significance of the SWIR channels (see Test 2 and Test 3 in Table 3, as well as Section 4.2). All these classification experiments mentioned in Table 3 were implemented with MATLAB (R2017).

## 3. Results

### 3.1. Comparison of Sensor Characteristics

Compared with other sensors, the S2A MSI had visible channels with a corresponding narrower spectral width, e.g., the green and red channels (Figure 2). The S2A MSI contained a relatively small part of the GF-1/-6 WFVs, whereas the GF-1/-6 WFVs covered most of (or even totally) the S2A MSI (Figure 3). At the same time, the L8 OLI and the S2A MSI mutually shared large parts in SRF in the blue and red channels, while showing obvious between-sensor discrepancy in the green channel (Figures 2 and 3). The S2A MSI had two NIR channels, of which the narrower one (i.e., 8a NIR) was comparable with the L8 OLI NIR, whereas the broader one presented overlay covering large parts with the GF-1/-6 WFVs. Generally, the GF-6 WFV RE channels were relatively broader and contained most parts of the S2A MSI RE channels (i.e., in RE1 and RE2) (Figures 2 and 3). The RE3 of the S2A MSI sharing small parts with the GF-6 WFV in the spectral region was not included in further investigation. Over the SWIR region, both the S2A MSI and the L8 OLI had two channels, which shared large overlap in SRF and showed between-sensor comparability. The percentage overlap in SRF is presented in Figure 3. Within the visible region, discrepancies were more obvious in the red and green channels, relative to the differences in the blue channel. For the red channel, about 70% of the GF-1 WFV was included in the GF-6 WFV, while small parts of the GF-1 WFV (less than 50%) were covered by the L8 OLI (or the S2A MSI). In contrast, the GF-1 WFV contained more than 95% of other sensors covering red wavelengths. On average, the GF-1 WFV and the GF-6 WFV were quite comparable in terms of the blue, green, and NIR channels, whereas they presented differences in the red channel.

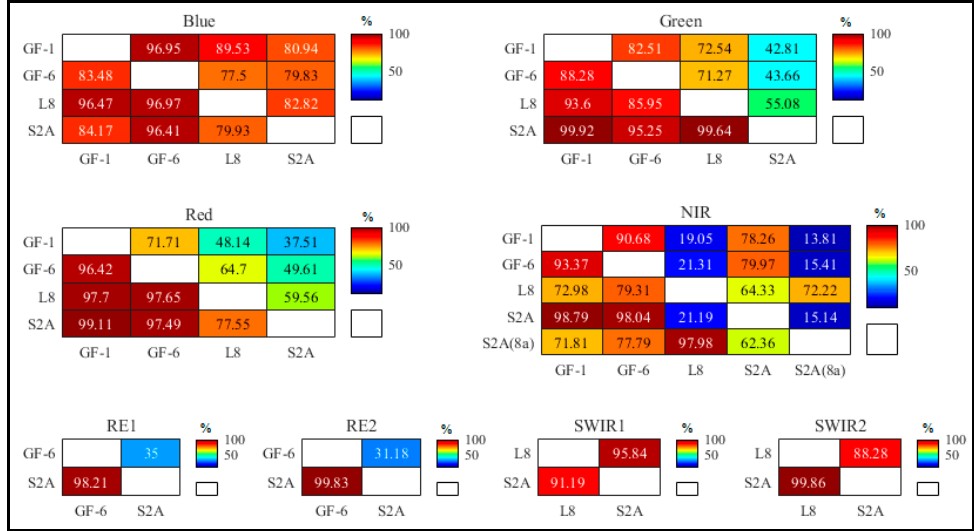

**Figure 3.** The percentage overlap (%) in SRF of corresponding (nominally equivalent) channels among the four sensors: the GF-1 WFV (GF-1), the GF-6 WFV (GF-6), the L8 OLI (L8), and the S2A MSI (S2A). In each subplot, the value at (A, B) measures the percentage overlap in SRF for sensor A (listed in left column) in sensor B (listed in bottom row), for a specific channel. For example, the value at (GF-6, L8) (being 64.7%) over red presents the percentage overlap in SRF for the GF-6 WFV in the L8 OLI, which is also the percentage for the part of the GF-6 WFV contained by the L8 OLI.

Generally, the between-sensor difference in channel reflectance was less obvious in the blue and green channels, with the MdRD being within (or approximate to) ±1.00% for all pairs (Figure 4). In the red channel, the GF-6 WFV presented larger MdRD relative to other sensors. Due mainly to their narrowness in NIR wavelengths, the L8 OLI and the 8a NIR (of the S2A MSI) differed significantly from other NIR channels, showing positive MdRDs of about 10.00%. At the same time, the broader NIR channel of the S2A MSI was more consistent with the GF-1/-6 WFVs NIR channel, whereas it had negative MdRDs against the narrower NIR channels. Slight differences in channel reflectance were observed (with MdRDs within ±1.00%) between the L8 OLI and the S2A MSI in the two SWIR channels.

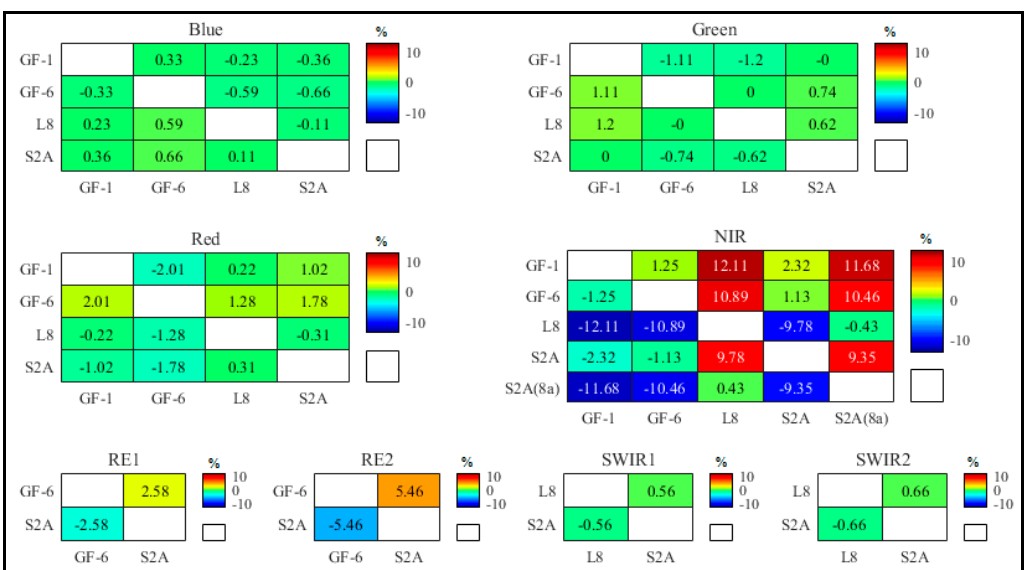

**Figure 4.** Between-sensor median relative difference (MdRD, %) in channel reflectance, for the blue, green, red, NIR, RE1, RE2, SWIR1, and SWIR2 channels. In each subplot, MdRD (A, B) measures the median relative difference in reflectance for a specific channel between sensor A (or channel, listed in left column) and sensor B (or channel, listed in bottom row), using sensor A as reference. For example, MdRD (L8, GF-6) presents the between-sensor difference for the GF-6 WFV using the L8 OLI as reference, while MdRD (GF-6, L8) indicates the bias for the L8 OLI using the GF-6 WFV as reference. Two NIR channels of the S2A MSI are included, with S2A(8a) indicating the 8a NIR (Table 1).

### 3.2. Methods Comparison

The MLC showed significantly varied performances under the four scenarios. Under a specific scenario with one sensor, compared to other classifiers, the MLC had relatively larger ranges in OA and Kappa coefficient (Figure 5). In contrast, the nonparametric classifiers showed relatively steady performances under each scenario, especially RF and KNN. With a small number of training samples (e.g., 50 or 100), the MLC showed approximate or even better performance compared to other classifiers including KNN, DT, and RF, being consistent with previous findings [1,45]. For the GF-1 WFV, under the scenario with 50 training samples, MLC performed without significant difference to these classifiers, whereas SVM significantly outperformed them. This demonstrated the advantage of the nonparametric classifiers with larger samples in training (e.g., 200 or 300), in terms of both accuracy and robustness. Compared with DT, RF as an ensemble classifier performed more robustly and obtained classification with higher accuracy, suggesting the effectiveness of the ensemble strategy. Meanwhile, under these scenarios, all paired classifiers with each sensor, except MLC and DT with L8 OLI, were different at the 5% significance level. With channel reflectance, the SVM with Gaussian kernel generally outperformed other classifiers in pixel-based classification.

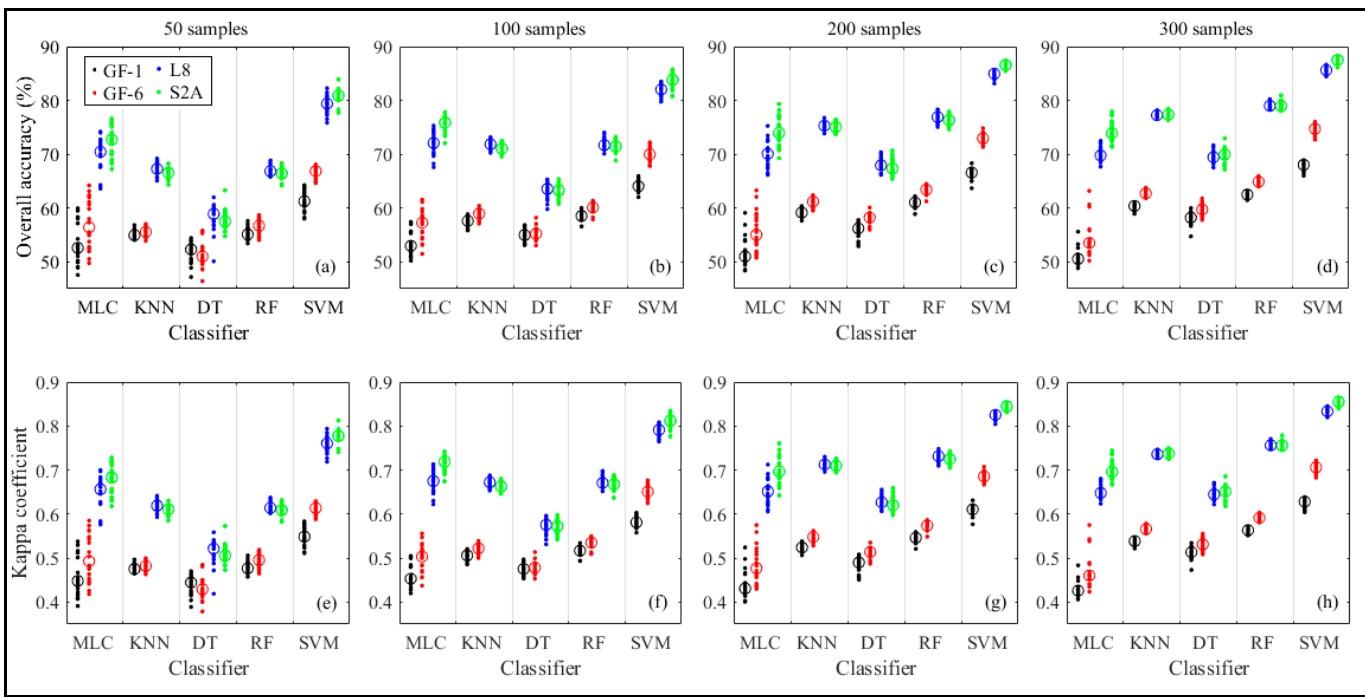

**Figure 5.** General assessment of classification results with different training samples using five classifiers separately for four multispectral sensors: the GF-1 WFV (with black), the GF-6 WFV (with red), the L8 OLI (with blue), and the S2A MSI (with green). The overall accuracy and Kappa coefficients are presented for cases under four scenarios: with 50 training samples (**a,e**), with 100 training samples (**b,f**), with 200 training samples (**c,g**), and with 300 training samples (**d,h**). In this figure, the dots show the assessment results of individual cases, while the circles indicate the median values. The classification results in this figure are those for Test 1 in Table 3, with a total of 1600 cases being conducted and presented in this figure.

### 3.3. Comparisons for Individual Classes

In addition, for the cases with MLC, RF, and SVM under the scenario with 200 training samples, individual classes were compared in terms of UA, PA, and the harmonic average (Figure 6). It was clear that all classifiers demonstrated unvaried performances in obtaining Gt and Ws from the multispectral reflectance, with stable and high accuracy no matter which sensor was considered. For most classes except Gt and Ws, with these classifiers, the L8 OLI and the S2A MSI had greater accuracy than the GF-1 WFV and the GF-6 WFV. In particular, for Sc and Sn, with MLC, the S2A MSI obtained significant differences in the median UA relative to the GF-1 WFV (or the GF-6 WFV), with approximate increases of 50% and 25% respectively. Relative to the GF-1 WFV (or the GF-6 WFV), through RF, the S2A MSI produced Sc, with the increases being about 23%, 17%, and 20% in the median UA, the median PA, and the median harmonic average, respectively. It also generated similar improvements for Sn (Figure 6).

Compared with MLC, both SVM and RF demonstrated an overall advantage in accuracy improvements and robustness for most individual classes. In particular, with the MLC, the accuracy of Sm varied significantly among different cases no matter which sensor was considered, whereas the RF performed steadily and produced Sm with greater accuracy. In terms of the harmonic average, the median increase for Sm generated through RF was about 28% with GF-1 WFV, whereas it was about 12% with the S2A MSI. However, for Cn and Cm obtained from the GF-1 WFV, the median increases in harmonic average were within 1–12% through RF. Generally, for individual classes, SVM outperformed RF, especially for Cn, Cm, and Sc (Figure 6). Furthermore, with RF (or SVM), the impacts of randomized training samples on classification (even for all individual types) decreased to some extent, although variations in accuracy were still observable for several classes (Figure 6).

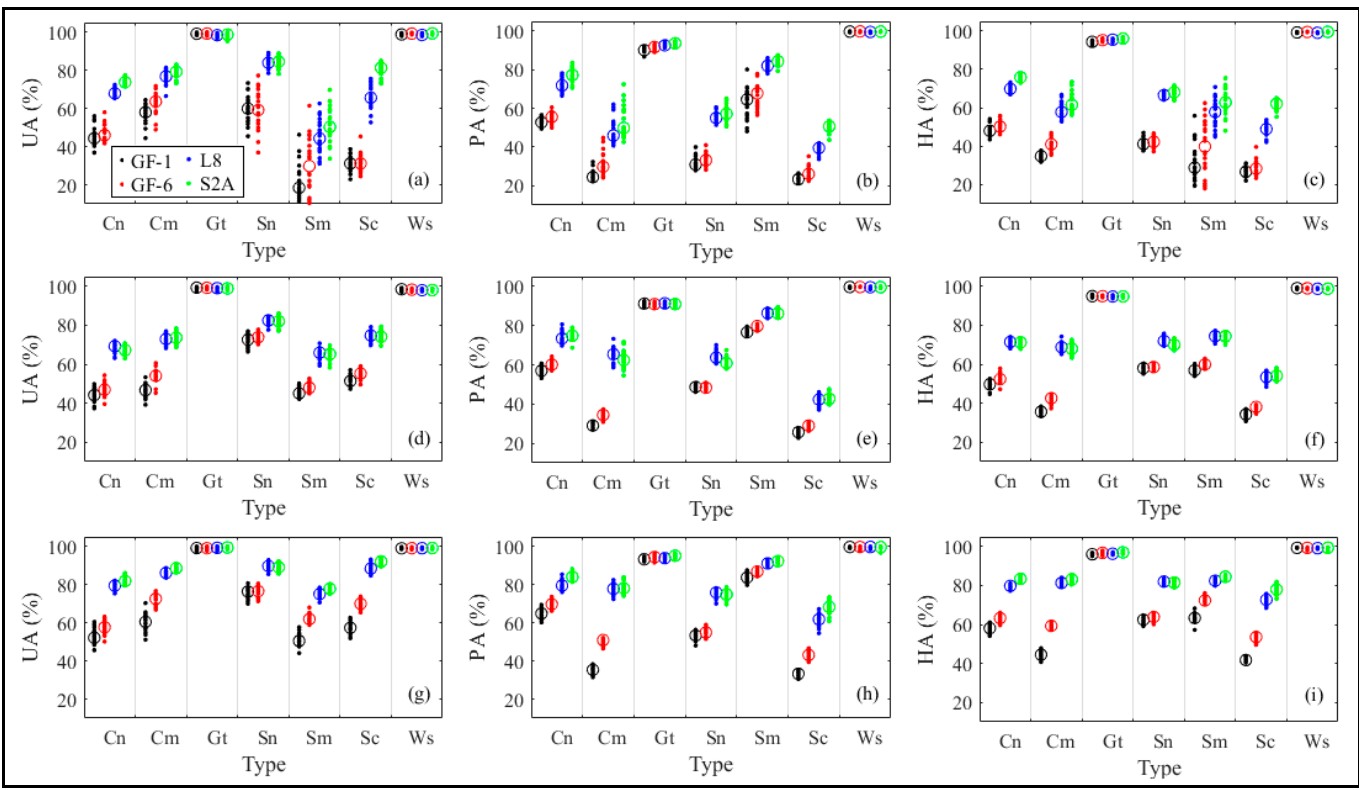

**Figure 6.** Classification results of 20 cases with MLC (upper), RF (middle), and SVM (bottom), under the scenario with 200 training samples: (**a**,**d**,**g**) user's accuracy (UA); (**b**,**e**,**h**) producer's accuracy (PA); (**c**,**f**,**i**) harmonic average (HA). Individual classes are corn-notill (Cn), corn-mintill (Cm), grass-trees (Gt), soybean-notill (Sn), soybean-mintill (Sm), soybean-clean (Sc), and woods (Ws). In this figure, the dots show the assessment results of individual cases, while the circles indicate the median values.

*3.4. Comparisons for Representative Cases*

　　Under the scenario with 200 training samples, with the various methods (i.e., MLC, RF, and SVM), each sensor generally performed well for Ws and Gt (Figure 6), but with slight misclassification of Ws into Gt (Figure 7). From the representative cases in Figure 7, both RF and SVM outperformed MLC for all sensors overall. In particular, with SVM, the increases in OA were about 15%. With RF, the GF-1 WFV and the GF-6 WFV both had great increases (about 10%) in OA, while the L8 OLI had a 7% increase relative to using MLC. For example, through RF, improvements for the GF-1 WFV were obvious for Sm with increases in both PA and UA (Figure 5), while reducing its omission associated with misclassification into Cm and Sn (Figure 7a,e). With MLC, the S2A MSI showed the best performance, with OA being about 22% and 19% greater compared to the GF-1 WFV and the GF-6 WFV, respectively. Meanwhile, with RF, the L8 OLI outperformed the GF-1/-6 WFVs, having significant OA increases (about 15%). On average, relative to the GF-1 WFV, the GF-6 WFV obtained classification with slight increases in OA, especially with SVM. Compared with the L8 OLI (or the S2A MSI), the GF-1 WFV had significant classification errors, especially for Cm (Figures 6 and 7). However, even in the best classification generated from the S2A MSI with SVM (Figure 7l), observable errors were still there for several classes. With reference to the ground truth, omission errors were significant for Sc (thus, with low PA), related to the misclassification of Sc into Cn, whereas its commission errors mainly resulted from misclassifying Sn into Sc. At the same time, for Sm, the observed omission errors and commission errors were mainly caused by its confusion with Cn, which was produced with relatively low UA due to the misclassification of Cm (Figure 6).

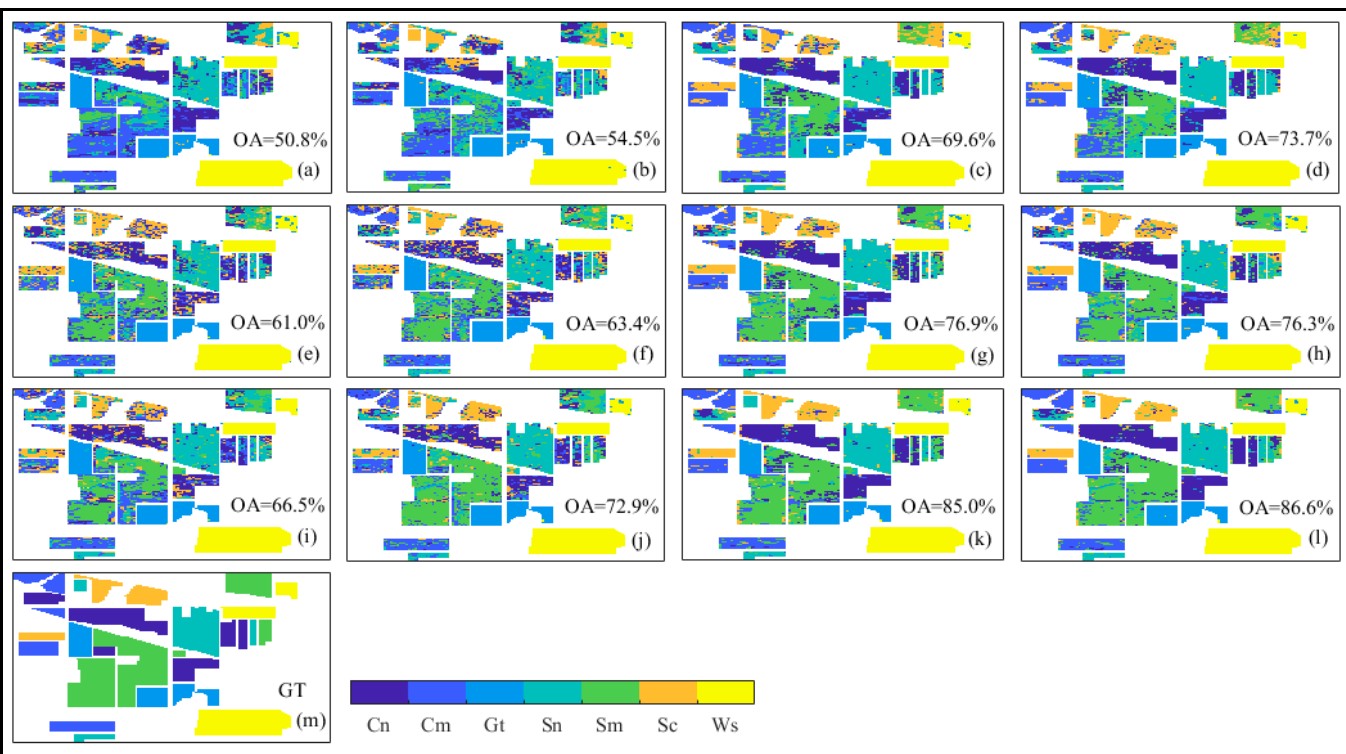

**Figure 7.** Representative classification results of different sensors with MLC, RF, and SVM under the scenario with 200 training samples: (**a**,**e**,**i**) GF-1 WFV with MLC, RF, and SVM, respectively; (**b**,**f**,**j**) GF-6 WFV with MLC, RF, and SVM, respectively; (**c**,**g**,**k**) L8 OLI with MLC, RF, and SVM, respectively; (**d**,**h**,**l**) S2A MSI with MLC, RF, and SVM, respectively; (**m**) corresponding ground truth (GT). In this figure, the representative case is provided with OA equal (or approximate) to the median OA of corresponding cases, representing the average performance of the cases with a specific sensor and a classifier.

## 4. Discussion

### 4.1. Classification Improvements through Observations vs. Other Methods

Advanced observation with more spectral channels, especially over the SWIR region, improved the accuracy of all individual classes to some extent (Figure 6), thereby generating more accurate classification overall (Figure 5). Moreover, using proper methods, classification results could be improved overall. Compared to MLC, RF and SVM had greater overall accuracy and showed more stable performance for most individual classes (Figures 5 and 6). Generally, the best classification results were produced using a proper classifier with a more advanced sensor (e.g., SVM with the S2A MSI in this investigation; see Figures 5 and 7). However, considering a specific sensor, for some classes, the improvements using only the classifier were limited. As shown in Figure 6, with the GF-1 WFV (or the GF-6 WFV), classes other than Ws and Gt were produced with low accuracy, although the improvements through SVM (or RF) were observed in most classes. This means that, for pixel-based classification, the performance of a specific classifier is largely affected or constrained by the sensor (or observation), as well as by the surface types to be retrieved.

Since the performance of the classifier varies with several factors, attention should be paid to method comparison and selection. Compared with MLC, RF as a nonparametric method with an ensemble strategy, performed steadily with greater accuracy under the scenario with a relatively large number of training samples (Figure 5). However, when a small number of samples (i.e., 50) were used for training, MLC outperformed RF overall, especially for the L8 OLI and the S2A MSI, but performance differed by case. Inconsistent conclusions among case studies in classifier comparison are possibly related to the training samples used (e.g., quantity and quality). Both SVM and RF showed the capability to

improve the classification for the GF-1 WFV and the GF-6 WFV to some extent, which are provided with imperfect channel settings (i.e., the position and number of channels). Accordingly, even without channels over the SWIR region, with SVM (or RF), the GF-1 WFV had an OA greater than 60%, whereas a difference of about 10% was observed with MLC. Since MLC was more sensitive to spectral inputs, significant overall improvements were obtained with more channels (e.g., the S2A MSI vs. the GF-1 WFV) (Figures 5 and 6).

Lastly, the properties (i.e., spectral) of classes intended to be extracted should be considered, in discussing the capability of a specific sensor or comparing sensors for land surface classification. Among the classes discussed, two types (i.e., Ws and Gt) were extracted with high accuracy regardless of the classifier, which also showed negligible between-sensor differences (Figure 6). For other classes, there were observable differences in accuracy between the GF-1/-6 WFVs and the S2A MSI (or the L8 OLI). Generally, a type with distinctly spectral characteristics would be easily classified against others, whereas confusion in the spectral signature among types would likely result in misclassification [29]. Four sensors had lower accuracy in Sc relative to other classes (Figure 6).

*4.2. Contribution of Spectral Channels to Classification*

The median values of normalized importance through RF tests for all cases under different scenarios are presented in Figure 8, investigating the relative contribution of individual channels in classification. For the GF-1 WFV, the NIR channel had the most significance in classification, whereas the green channel was less important relative to other visible channels. For the GF-6 WFV, its NIR channel showed the most significance, followed by the RE2, except with 50 training samples. Meanwhile, the RE1 shared similar importance with other channels over the visible spectral region. For the L8 OLI, its NIR channel (with a narrow spectral range) had the greatest contribution, and the two SWIR channels were more important relative to all visible channels. Furthermore, the SWIR1 channel was generally greater in normalized importance score than the SWIR2 channel. For the S2A MSI, its SWIR1 channel made the greatest contribution to classification. Meanwhile, the NIR channels (i.e., NIR and 8a NIR) and the RE2 channel, as well as the SWIR2 channel, were generally more important than the visible channels under scenarios with a relatively larger number of training samples (Figure 8), whereas the RE1 had less impact. Without channels over the SWIR region, both the GF-1 WFV and the GF-6 WFV showed inferior performance (Figures 6 and 7).

In fact, the reflectivity over the SWIR region is sensitive to vegetation water content and the structure of leaves. In particular, the normalized difference moisture index (NDMI), which embeds the SWIR spectral information, is used to detect moisture levels in vegetation [46]. As a reliable indicator of water stress, the NDMI is more sensitive to vegetation disturbances and more resistant to data noise than several other vegetation indices (e.g., the normalized difference vegetation index) [46–48]. The SWIR spectral information also enhances the features of water bodies, e.g., in the modified normalized difference water index [49]. For classification, the effectiveness of the SWIR data, as well as the NIR (i.e., 8a NIR) data, of the S2A MSI was verified in distinguishing crop types [50]. In the tree species classification over tropical forests, the SWIR channel (i.e., of WorldView-3) contributed to an increase (up to 7.8%) in average accuracy [51]. Due mainly to its lack of SWIR channels, the GF-6 WFV was inferior to the S2 MSI in mapping carbonate rocks [52]. At the same time, simulation tests illustrated obvious improvements in classification upon adding the SWIR and RE channels compared to the results only based on visible and NIR channels [53].

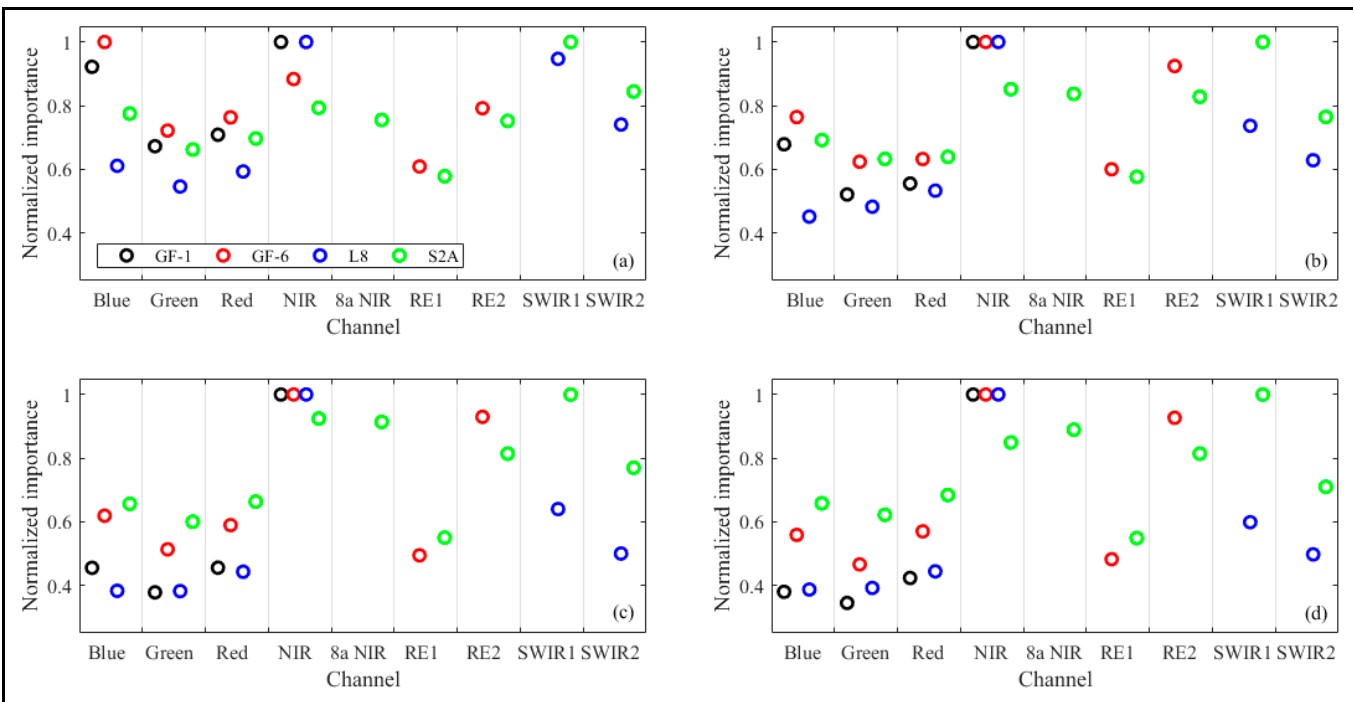

**Figure 8.** Normalized importance of individual channels in classification with different training samples through RF tests: (**a**) for cases with 50 training samples; (**b**) for cases with 100 training samples; (**c**) for cases with 200 training samples; (**d**) for cases with 300 training samples. The median values of normalized importance through RF tests for all cases under individual scenarios are presented. Information on the channels used in classification is provided in Table 1 and Figure 2.

Two types of experiment were further investigated separately to understand the significance of channels over the SWIR region in classification. Specifically, the approaches involved (1) simply adding the SWIR channels of the L8 OLI to the GF-1 WFV, i.e., adding SWIR1, adding SWIR2, or adding SWIR1 and SWIR2 together (Test 2 in Table 3), and (2) reducing the SWIR channels of the L8 OLI (Test 3 in Table 3). The artificial channel combination (or reduction) was compared with the original channel settings in terms of overall accuracy and Kappa coefficient. Figures 9 and 10 present the comparative analyses, in which the training samples and validation samples used previously (Section 2.4) were implemented for individual cases. Relative to other classifiers, upon adding the SWIR channels (of the L8 OLI) to the GF-1 WFV, the MLC produced a greater increase in accuracy (Figure 9). Compared with SWIR2, SWIR1 made a greater contribution to the improvement in classification, consistent with the findings from the importance tests (Figure 8). Upon adding both SWIR channels, the greatest improvement was observed. In these cases, the original L8 OLI had greater accuracy compared to the L8 OLI without SWIR channels (Figure 10). Specifically, according to RF, the differences were about 15% and 0.18 in overall accuracy and Kappa coefficient, respectively. Nevertheless, compared to the L8 OLI without SWIR channels, the GF-1 WFV showed equivalent performance overall. In particular, with MLC, the GF-1 WFV even slightly outperformed the L8 OLI without SWIR channels.

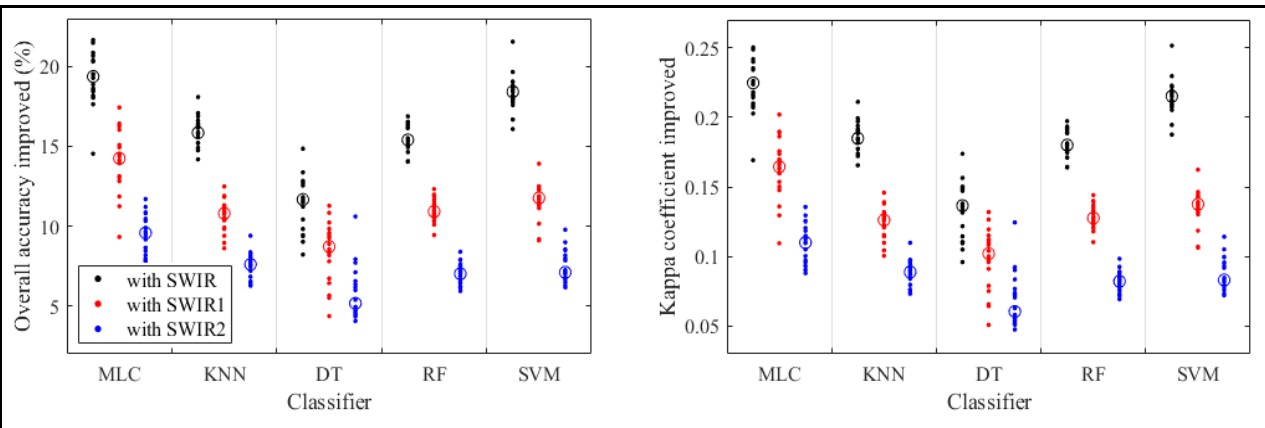

**Figure 9.** Improved performance of different channel combinations compared to the original GF-1 WFV considering 20 cases under the scenario with 200 training samples for each class: upon adding SWIR1 (red), upon adding the SWIR2 (blue), and upon adding both SWIR channels (black) of the L8 OLI. In this figure, the dots show the improved performances of individual cases, while the circles indicate the corresponding median improvements.

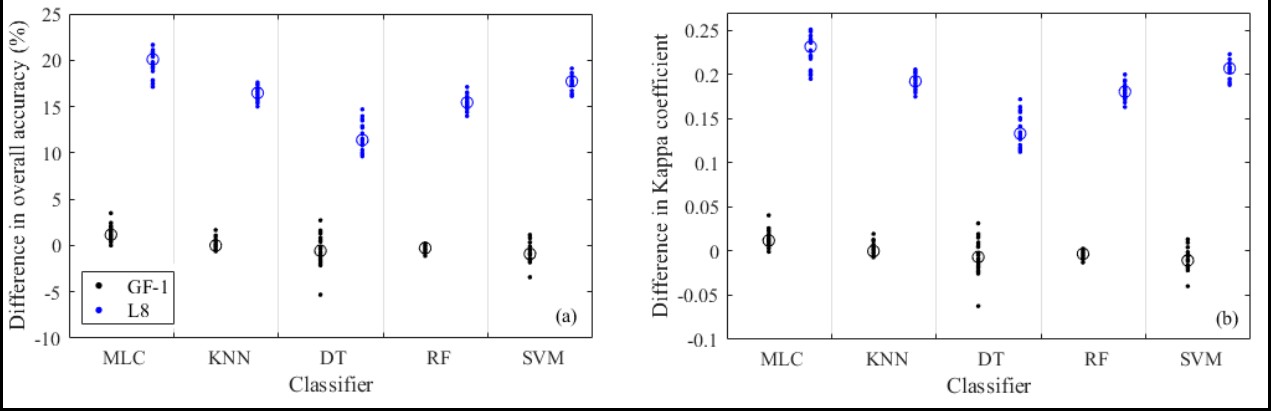

**Figure 10.** Comparisons of the original L8 OLI (blue) and the GF-1 WFV (black) to the L8 OLI without both SWIR channels: (**a**) difference in overall accuracy; (**b**) difference in Kappa coefficient. The difference was calculated using the result from the L8 OLI without both SWIR channels as the baseline. Similar to Figure 9, 20 cases under the scenario with 200 training samples for each class were considered. In this figure, the dots show the differences for individual cases, while the circles indicate the corresponding median difference.

### 4.3. Parameter Determination in Performing RF

TreeBagger, as an RF model embedded in MATLAB (R2017), was trained and adopted in this study. To perform TreeBagger, two parameters are required: the number of decision trees and the number of leaves. Therefore, series of tests to determine suitable parameters were performed before RF application. Generally, the mean squared error (MSE) declined sharply upon increasing the number of trees to 50, which subsequently stabilized regardless of the number of leaves (Figure 11). Meanwhile, considering a specific number of trees, a greater MSE was possibly obtained with a larger number of leaves, and the minimum MSE was observed with one leaf or three leaves. With the same parameters, compared to the GF-1 WFV, the S2A MSI achieved a lower MSE. Thus, in all cases with RF, the number of decision trees was set to 100 while the number of leaves was three. In previous studies, different parameters were set to perform RF for classification. In terms of the number of trees, it was 200 in [7] and 100 in [16]. After a series of tests, the best classification with RF was obtained when the number of trees was 60 using four channels or 20 using

six channels [45]. Although the number of trees suitable for classification is case-specific, RF with at least 100 trees is preferred [31]. In terms of the number of leaves (random variables available at each node), it was set as one in [45], whereas it was determined as the square root of the number of inputs in [16]. An applicable approach to determine the number of leaves through tests was recommended in [31], although it generally had a moderate impact on classification accuracy. In our experiments, the number of decision trees and the number of leaves were determined on the basis of tests, the results of which were generally consistent with previous investigations [16,31]. Furthermore, with these settings, the RF performed steadily regardless of sensors, as shown previously (Figures 5 and 6). Thanks to RF, the effects of imperfect channel settings (e.g., the position and number of channels for the GF-1 WFV) were eliminated to some extent. However, the influence associated with the parameters on final classification was not discussed in detail, which is necessary for the application of RF. As mentioned before, considering possible effects associated with sample imbalance, seven classes with larger samples were mainly considered for classification comparison. The impacts of the data characteristics in training should be further investigated.

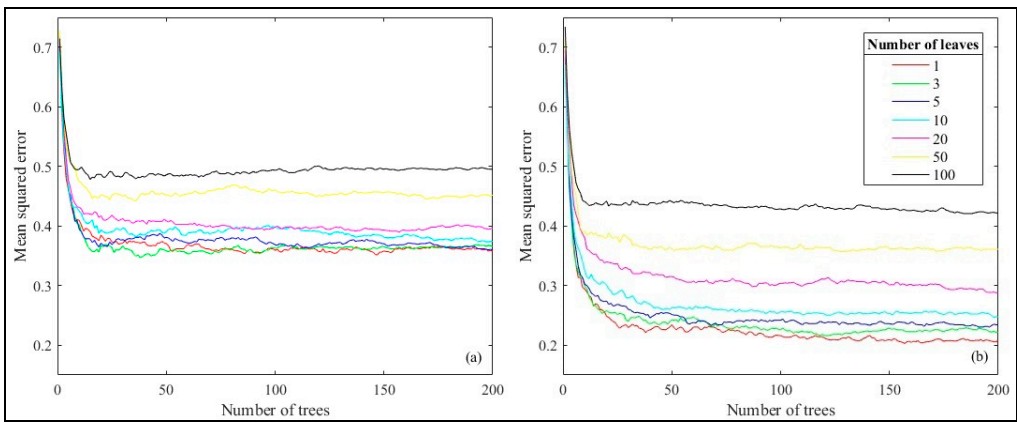

**Figure 11.** Pretraining tests for parameters determination of RF: (**a**) for the GF-1 WFV; (**b**) for the S2A MSI.

The simulated multispectral reflectance from AVIRIS hyperspectral imagery was used in the experiments to investigate the between-sensor comparability among the four sensors. In addition to spectral characteristics, other aspects of the multispectral imagery need to be discussed for data application in practice, including spatial resolution, radiometric properties (e.g., dynamic range, radiance quantization, and signal-to-noise ratio), and swath width. Meanwhile, with specific imagery, choices in data processing affect the final classification to some extent, such as the feature combination and classifiers [21]. Spatial resolution is considered important to surface mapping [1,54]. For specific multispectral imagery, limited spectral richness can be compensated for to some extent with a higher spatial resolution [55,56]. For example, with 16 m, the GF-1/-6 WFVs demonstrated a relative advantage for specific classes, as well as for vegetation coverage, compared to the L8 OLI [13,15,57], compensating for its lack of spectral channels over the SWIR region. A sensor's revisit time is also valuable for specific applications (e.g., disaster assessment, change detection, and crop growth monitoring). With advantages in terms of both spatial resolution and revisit time, the combination of multitemporal GF-6 WFV observations showed similar performance to the S2 MSI data [19]. Considering radiometric properties, inconsistent findings were presented in several case studies [58–60]. In simulation tests of the L8 OLI from the Indian Pines AVIRIS data, radiance quantization had more influence over the SWIR region [35]. This suggests that the quantization effects can largely be associated with observation and land surface characteristics. For example, for glacier surface classification, the dynamic range in radiance was the most important feature [58]. Furthermore, considering the improvements in land-use/cover classification from remotely

sensed observation, different approaches are possibly effective [29], e.g., by incorporating vegetation indices.

## 5. Conclusions

This paper firstly compared the spectral characteristics (e.g., spectral region and spectral response function) of the GF-1 WFV, the GF-6 WFV, the L8 OLI, and the S2A MSI. Additionally, the channel reflectance of individual sensors was simulated from the Indian Pines AVIRIS data. Consequently, using the simulated channel reflectance, comparisons in classification were made under different scenarios, in which five classifiers (i.e., MLC, DT, KNN, RF, and SVM) were implemented. Major conclusions from this study concerned four aspects:

- General comparison in sensor characteristics and in channel reflectance. On average, the GF-1 WFV was quite comparable to the GF-6 WFV in all visible and NIR channels, except the red channel. Over the visible region, the L8 OLI showed consistency with the S2A MSI in the blue and red channels, but differed obviously in the green channel. Both the S2A MSI and the L8 OLI have two SWIR channels, showing between-sensor comparability.
- General comparison in classification. Generally, the GF-1 WFV was inferior to other sensors in classification, and the GF-6 WFV slightly outperformed it. Generally, the differences in classification between the L8 OLI and the S2A MSI varied with the classifier, although similar performances were observed for the two sensors. With channel reflectance, SVM outperformed other classifiers in pixel-based classification, likely addressing the problems related to imperfect channel settings (i.e., the position and number of channels). Since several factors possibly influence the performance of a classifier, attention should be paid to method selection.
- Importance of channels for classification. For the GF-1/-6 WFVs, the NIR channel had the greatest importance in most classification cases. Meanwhile, for the L8 OLI and the S2A MSI, the NIR and SWIR channels were more significant relative to other channels. For the L8 OLI, SWIR1 (1560–1660 nm) was more effective than SWIR2 (2100–2300 nm). Without channels over the SWIR region, both the GF-1 WFV and the GF-6 WFV showed inferior performance.
- Merits of the simulated data collection on the basis of which more tests were conducted. The multispectral channel reflectance simulated from the well-processed hyperspectral data was used, to focus mainly on the differences among sensors in spectral characteristics. In addition, with ground truth, comparative analyses among sensors and among classifiers were readily made. However, the spatial coverage of the Indian Pines AVIRIS data was relatively small.

Channel reflectance was only used in classification experiments to understand the significance of individual channels. Nevertheless, adding spectral indices and combining multitemporal observations [19] can be considered to improve the accuracy in practice. The results verified the significance of channels over the SWIR region for classification. Sensors (e.g., the GF-1 WFV) without the SWIR channels generally performed inferiorly for most classes. This suggests possible challenges for change detection or monitoring surface dynamics using different sensors jointly (e.g., sensors with and without SWIR channels). To enhance the capability of the GF-1/-6 WFVs in classification, other effective methods are necessary, although combing multitemporal observations is valuable. In addition to the spectral channel, other factors that possibly influence classification were not investigated in this study, such as spatial resolution, radiometric quantization, and temporal resolution. Furthermore, since the simulated multispectral reflectance from the Indian Pines AVIRIS data was used, this paper was dependent on limited spatial coverage. Further research on this issue is definitely required, to investigate and improve the comparability in classification among different sensors.

**Author Contributions:** Conceptualization, F.C. and C.W.; formal analysis, F.C., W.Z. and Y.S.; funding acquisition, F.C., Y.S. and C.W.; methodology, F.C., W.Z. and C.W.; project administration, F.C. and C.W.; writing—original draft, F.C., W.Z. and C.W.; writing—review and editing, W.Z., Y.S., L.L. and C.W. All authors have read and agreed to the published version of the manuscript.

**Funding:** This research was funded by the Fujian Natural Science Foundation, China (No. 2021J011190), the High-Level Talents Research Project of Xiamen University of Technology (No. 4010520004), the Fujian Educational Research Projects of Young and Middle-Aged Teachers (No. JAT200453), the National Natural Science Foundation of China (No. 42101250), and the Water Conservancy Science and Technology Project of Jiangxi Province (No. 202224ZDKT11 and No. 202123YBKT16).

**Data Availability Statement:** The data supporting the findings in this study are openly available from the Purdue University Research Repository at doi: 10.4231/R7RX991C, accessed on 10 January 2023, reference number [30].

**Acknowledgments:** The authors sincerely appreciate CRESDA for the provision of the spectral response functions of the GF-1 WFV and the GF-6 WFV, as well as NASA and ESA for the provision of the spectral response functions of the Landsat 8 OLI and the Sentinel-2 MSI, respectively. In addition, the Indian Pines AVIRIS calibrated data and associated ground-truth records were accessed through the Purdue University Research Repository.

**Conflicts of Interest:** The authors declare no conflict of interest.

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
