# Peer review of "Comparison of Simulated Multispectral Reflectance among Four Sensors in Land Cover Classification"

_remotesensing, doi:10.3390/rs15092373_

Round 1

Reviewer 1 Report (Previous Reviewer 2)

In the revised version, the Authors performed several revisions to address my concerns. However, these several revisions satisfied my important but minor importance concerns, but the main and most significant problem still stays in the paper. 

To be clear, classification performance is not only a function of spectral properties but also of the radiometric, temporal, and SPATIAL resolution (GSD) of the images. The authors used AVIRIS hyperspectral data to simulate other sensor data. However, Sentinel 2 has both 10 and 20m spatial resolution bands, Landsat has 30m resolution bands and GF has 16m bands.

Thus, such experimental design has an important limitation. In addition, the region covered by the AVIRIS data is quite limited and includes few LULC classes, which again limits the reproducibility of the results.

Author Response

Thanks. We sincerely appreciate the comments and suggestions proposed by the reviewer. In this paper, impacts of spectral channel (e.g., number of channels, spectral range of the channel, and spectral response function) were mainly discussed on use/cover classification. Meanwhile, the comparability issues were investigated among four sensors (i.e. GF-1 WFV, GF-6 WFV, L8 OLI, and S2A MSI) of which the multispectral imagery is accessible free of charge. Investigations were mainly based on the simulated multispectral reflectance from hyperspectral records (i.e. the Indian Pines AVIRIS data), while the between-sensor differences among the four sensors in number of channels, spectral range of the channel, and spectral response function were considered. However, as mentioned by the reviewer, there are many factors affecting classification results and accuracy in practice. For example, the advantages of the GF-1/6 WFV in spatial resolution and revisit period possibly compensate their shortages in spectral channel [1].

To our knowledge, it has been the first paper discussing the comparability issues among the four sensors simultaneously. Findings of this paper show readers the possible inconsistency problem among the four sensors. We believe findings and conclusions in this paper are generally applicable, which showed accordance with several previous studies. However, more case studies should be carried out further with real images over areas with different geographical and environmental backgrounds. Accordingly, limitations of this study has been added in “5.Conclusions”.

[1] Xia, T.; He, Z.; Cai, Z.W.; Wang, C.; Wang, W.J.; Wang, J.Y.; Hu, Q.; Song, Q. Exploring the potential of Chinese GF-6 images for crop mapping in regions with complex agricultural landscapes. Int. J. Appl. Earth Obs. 2022, 107, 102702.

Reviewer 2 Report (New Reviewer)

The comparison in this study is purely based on simulated reflectance data and using a very small sample area.

In real world applications the relative merit of each sensor will be affected by other factors such as:  spatial and temporal resolution of the imagery, point spread function of the sensor, signal to noise ratio, observation times, sensor view angles etc. There will also be impacts from different pre-processing applied by the distributors of these data. The combination these factors could lead to differences that are significantly greater than spectral differences.

The very small study area also a limitation as classification of a larger area is likely to have more spectral variation and variation in view angle, atmospheric effects etc. This very small study area is unlikely to be sufficient to characterize spectral differences between the sensors for mapping applications so the classification results may not be representative of operational applications.

The authors have partially covered the above limitations in line 612+ but it needs to be more explicit.

-          The introduction needs to review the impact of these other differences in sensors based on published literature

-          The abstract and conclusions need to clearly state the limitations of this being purely based on simulated data and won’t reflect the real-world performance of these sensors in operational applications

At line 84 you state “paired observations from four sensors was not available”, which is why you based the study on simulated reflectance.  Perhaps you could compare data for each sensor against each other sensor. This only requires pairs of observations not four near simultaneous observations and would provide more meaningful results.

Author Response

Thanks a lot for these comments and suggestions. In practice, there are many factors, which affect final land use/cover classification from remotely sensed imagery, the spectral channel among others. This paper has been the first one discussing the comparability issues among the four sensors (i.e. GF-1 WFV, GF-6 WFV, L8 OLI, and S2A MSI) simultaneously, mainly in land use/cover classification based on the multispectral reflectance simulated from hyperspectral records (i.e. the Indian Pines AVIRIS data). Accordingly, the between-sensor differences in spectral channel (e.g., including the number of channels, spectral range of the channel, and spectral response function) and their impacts on classification were mainly investigated. At the same time, spatial coverage of the Indian Pines AVIRIS imagery is relatively small, although it has been widely used as benchmark in classification. By the way, findings of this paper show readers the possible inconsistency problem among the four sensors, and remind data users to pay attention to this inconsistency issue. However, more case studies should be carried out further with real images over areas with different geographical and environmental backgrounds. According to the reviewer’s suggestions, limitations of this study have been mentioned in both “Abstract” and “5. Conclusions”.

Furthermore, other related papers have been added in “1.Introduction” and “4.Discussion” correspondingly.

[1] Gong, P.; Howarth, P.J. An assessment of some factors influencing multispectral land-cover classification. Photogramm. Eng. Rem. S. 1990, 56(5), 597-603.

[2] Chouari, W. Contributions of multispectral images to the study of land cover in wet depressions of eastern Tunisia. Egypt. J. Remote Sens. 2021, 24, 443-451.

[3] Lassalle, G.; Ferreira, M.P.; Rosa, L.E.; Scafutto, R. D.; Filho, C.R. Advances in multi- and hyperspectral remote sensing of mangrove species: A synthesis and study case on airborne and multisource spaceborne imagery. ISPRS J. Photogramm. 2023, 195, 298-312.

[4] Xia, T.; He, Z.; Cai, Z.W.; Wang, C.; Wang, W.J.; Wang, J.Y.; Hu, Q.; Song, Q. Exploring the potential of Chinese GF-6 images for crop mapping in regions with complex agricultural landscapes. Int. J. Appl. Earth Obs. 2022, 107, 102702.

Round 2

Reviewer 1 Report (Previous Reviewer 2)

The newly added comments in Discussion and Conclusion parts are adequately describing the limitations of this work and limited test setup. I have no additional comments.

Author Response

Many thanks for the reviewer's  comments and suggestions during the peer review.  The authors will conduct other investigations on the between-sensor comparability issue with paired and real observations.  

This manuscript is a resubmission of an earlier submission. The following is a list of the peer review reports and author responses from that submission.

Round 1

Reviewer 1 Report

The authors addressed my  comments. The paper is  publishable now.

Reviewer 2 Report

The revised version of the manuscript provided important improvements in terms of the Introduction, evaluation design, and article flow. However,  important methodological gaps and a theoretical problem still exist in the paper.

Classification performance of the satellite image is greatly dependent on several related factors: the spatial resolution (GSD), radiometric response (or quality), spectral resolution and band design, whereas there are also application-based factors: algorithm selection, training sample quality and testing design (sampling schema for ground truth in accuracy assessment).

The authors mostly focused on spectral comparison with a simulated data construction for the interested satellite sensors from hyperspectral imagery on the sensor-related factor side. However, this focus brings in an important deficiency in their analysis as they did not consider the spatial resolution and radiometric quality (as they mentioned in lines 644 -646). Thus their findings will not exactly match the real data conditions, and thus will not match the real earth observation problems. Authors should consider that even if all the remaining parameters are kept identical, just the increase in spatial resolution will greatly affect the performance and accuracy of LULC classification, especially for complex and heterogeneous landscapes.

2.  the non-parametric ML-based algorithms selected in this research require high amount of training data, thus Authors findings about higher accuracy of maximum likelihood with low training data is an expected result (no- novelty here). However, another option for ML-based algorithm is the support vector machine, which is proven to provide higher accuracy when the training sample is limited. Although Authors neglected the use of SVM due to their references reporting better performance of RF, it will not be a convincing reason unless the performances of these two (SVM - RF) under limited training data conditions. 

Reviewer 3 Report

The authors conducted a comparative analysis of the four sensors in classification were made. As a result, the GF-1 WFV and GF-6 WFV were inferior to Landsat 8 OLI and Sentinel-2 MSI and there were no differences between OLI and MSI. The paper is on an interesting topic and the results are useful for the remote sensing community

There are some issues with the manuscript as the authors can find some of them in the following. 

1. Introduction

Why did you choose MLC, KNN and DT for comparisons?

Recently, some methods based on deep learning have been proposed and their high performances have been reported. More reviews on classification methods are required.

2.4 Comparisons in Classification

Could you provide the ground truth map?

You said you didn't you some methods based on deep learning due to the insensitiveness to the noise of the target data.

Did you have enough data for evaluation? Please provide authorities.

Your treatment and explanation of RF as a process is not sufficient. What are the impacts of training data characteristics (class imbalance and mislabeling) on the performance of RF?

Why didn't you use any vegetation indices? Some works reported that VIs were effective for improving accuracies.

3.2. Comparison in Classification

Did you conduct any test for evaluating differences?

Were the classification accuracies significant?

4. Discussion

You said PA, UA and F1-score were used in assessing individual classes. However, more information should be presented on the performance of the individual variables and how they correlate to classes. How exactly do you reach a conclusion based on them?